# When COVID-19 Is Not All: Femicide Conducted by a Murderer with a Narcissistic Personality “Masked” by a Brief Psychotic Disorder, with a Mini-Review

**DOI:** 10.3390/ijerph192214826

**Published:** 2022-11-11

**Authors:** Donato Morena, Nicola Di Fazio, Raffaele La Russa, Giuseppe Delogu, Paola Frati, Vittorio Fineschi, Stefano Ferracuti

**Affiliations:** 1Department of Anatomical, Histological, Forensic and Orthopaedic Sciences, Sapienza University of Rome, Viale Regina Elena 336, 00185 Rome, Italy; 2Department of Clinical and Experimental Medicine, University of Foggia, 71100 Foggia, Italy; 3Department of Human Neuroscience, Faculty of Medicine and Dentistry, Sapienza University of Rome, Piazzale Aldo Moro, 5, 00185 Rome, Italy

**Keywords:** femicide, crime, SARS-CoV-2, pandemic, COVID-19-related psychosis, brief acute psychotic disorders, Brief Psychotic Disorder, narcissistic personality, psychopathy, forensic psychiatry, risk management

## Abstract

Several cases of COVID-19-related mental disorders have emerged during the pandemic. In a case of femicide that occurred in Italy during the first phase of the pandemic, coinciding with a national lockdown, a discrepancy arose among forensic psychiatry experts, particularly toward the diagnosis of Brief Psychotic Disorder (BPD) related to COVID-19. We aimed to discuss the evaluation of the case through an integration of information and a literature review on comparable reported cases. An analysis of the diagnosis of brief acute psychosis was then performed, as well as a mini-review on cases of COVID-19-related psychosis. Results showed that psychotic symptomatology was characterized by polythematic delusions that always involved a SARS-CoV-2 infection. To a lesser extent, the delusions were accompanied by hallucinations, bizarre cognitive and associative alterations, insomnia, hyporexia, dysphoria, and suicidal behavior. No particularly violent acts with related injury or death of the victim were described. Finally, we could hypothesize that our case was better represented by a diagnosis of personality with predominantly narcissistic and partly psychopathic traits. The present case highlighted the importance, in the context of forensic psychiatry, of integrating assessments with the crime perpetrators, namely through accurate clinical interviews, neuropsychological tests, diachronic observations, and comparison with similar cases present in the literature. Such an integrated approach allows precise evaluation and reduces the odds of errors in a field, such as forensic psychiatry, where a diagnostic decision can be decisive in the judgment of criminal responsibility. Moreover, discerning forensics from health cases represents an important issue in risk management.

## 1. Introduction

The SARS-CoV-2 pandemic, ongoing since 2019, represents the most critical global crisis in recent years, due to its impact on a wide range of aspects of the life of all citizens [1,2]. The initial reaction to this unexpected event was characterized by profound dismay and incredulity, catching both society and national institutions unprepared. There have been many debates and political confrontations on how to deal with the emergency. However, in the initial phases of the epidemic, the dramatic increase in deaths caused by COVID-19 and the lack of effective treatments made it necessary to apply harsh health policy strategies such as lockdowns and quarantines. Excluded from these measures were only some categories of workers necessary for the functioning of essential services, such as healthcare personnel. This category was among the most exposed to the stress of the pandemic, forced to live with the dangers of contagion and with an overload of work resulting in burnout [3]. The decisions on the application of lockdowns have undoubtedly led to the indiscriminate isolation of people, with the risk that some evident or latent conflicts could explode into dangerous domestic offenses [4]. Actually, an increase in the episodes of Intimate Partner Violence and in the severity of physical assaults was recorded worldwide [5,6].

Italy was one of the main countries affected by the viral spread. The Lombardy region, in the north of the country, was the area in which there was the most rapid increase, between February and March 2020, of people dying because of the infection. In the city of Nembro, for example, more people died only in March 2020 than in the entire previous year or in any single year since 2012 [7]. The increase in deaths in such a short time was so dramatic that it required the intervention of the Italian army to transport the coffins from Bergamo to the crematoria outside the Lombardy region after they had accumulated, unburied, in the city’s cemetery [8]. Such dramatic scenes being observed on television had a profound effect of emotional disturbance on the population, who suddenly found themselves at risk for their life. In this context, in Italy, but in a region still relatively spared from viral circulation, Mr. K, a 27-year-old nurse at the time, was accused of killing his partner, a medical student close to graduation. Her death occurred at night, due to cardiorespiratory arrest by acute asphyxia from direct suffocation caused by compression on the face and neck. Mr. K, after having first reported the murder himself, initially pleaded guilty to the act, then reported having a “black-out” regarding the event. In the various interrogations, he often mentioned different and, in some cases, conflicting elements that led to the need for a psychiatric evaluation to determine his legal competence at the time of the events. This case is presented with the aim of discussing the diagnosis of Brief Psychotic Disorder (BPD), in the context of forensic psychiatry and a particular period such as of pandemic.

## 2. Materials and Methods

All relevant juridical and medico-legal material of the case that emerged during the public debates was examined. The evaluated person agreed to make his documents public upon acceptance of the forensic psychiatric assessments. All documents were attached to the judicial file in the public domain. The Court materials, once the case is decided by a Judge’s ruling, are subject to consultation, and the following data can be extracted. Constructs of Brief Psychotic Disorder (BPD), according to the DSM-5, and Acute and Transient Psychotic Disorder (ATPD), according to the ICD-11, were analyzed and compared. A mini-review on cases of COVID-19-related psychosis was then performed. The studies were selected based on the research topic, i.e., brief acute psychotic disorders present in the world’s acknowledged databases, such as Web of Science, PubMed, Scholar, and Scopus from the period from January 2020 to August 2022. Search terms were “(brief psychotic disorder OR BPD OR acute and transient psychotic disorder OR ATPD OR brief acute psychotic disorder OR psychotic disorder) AND (coronavirus OR COVID-19 OR COVID)”.

More relevance was therefore given to studies with a higher number of cases and information derived from structured clinical settings (hospitals, mental health services). The clinical features found in the selected studies were carefully evaluated to compare them with those that emerged in the present case, to establish a differential diagnosis. Finally, an explanatory hypothesis about the case was developed.

## 3. The Case of Mr. K

### 3.1. First Interview with Authorities

The first statements reported to the judicial authority by Mr. K consisted of the confession of the murder. He claimed that it was the result of a quarrel with his partner, which lasted from the previous evening until the early hours of the morning, due to his anxiety about the COVID-19 emergency in Italy. He stated that she was responsible for transmitting the virus to him and to his subsequently deceased relatives (which never happened) after she moved to infected areas. He declared that anger and anxiety over contracting the virus prompted him to kill his partner, hitting her with a knife in the abdomen and with a lamp in the head, strangling her with his hands, and putting his feet in her face.

### 3.2. Second Interview with Authorities

After about two months, at the second interrogation with the judicial authority, Mr. K added some elements to the previous statements. He reported that in the evenings before the events, he had repeatedly contacted his father who was in a different region from his residence. For several days, he had expressed his desire to return to his family, but his partner would always oppose this, giving him a sense of threat.

Anyway, during the interview he often contradicted himself. At first, he said that he had not quarreled with his partner that evening; then, he reiterated that they had quarreled over his anxiety related to the virus. He retracted the earlier statements, claiming that he only used his hands as a means of offense to strangle her. She had tried to counteract, but as he was “a man”, she had no chance. Later, before advising police, he had unsuccessfully attempted suicide through phlebotomy of the wrists and electrocution, dipping a hairdryer in the water of the bathtub. He reported having perceived both his life and that of his own family members to be threatened by the relatives of his partner, who had the intention of killing him. He did not know the reasons for such intentions. In the personal reconstruction of the day before the events, Mr. K reported that he went to work in the morning and returned home in the afternoon. He then noticed that his partner was contacting his father and her father by telephone, reporting that he was planning to escape. Failing to understand the reason for these calls, he contacted his father himself, who begged him to do whatever his partner said. From these sentences of his father, he understood that “something was wrong”. Things then got worse when his partner told him not to go to work anymore. Mr. K denied the possibility that this could have been a form of precaution motivated by the belief that his job as a nurse was probably causing him a lot of stress. He described the relationship with his partner as a “normal relationship”, quite peaceful except for some quarrels “like those of any couple”. Both were jealous of each other and had the habit of checking each other’s cell phones.

### 3.3. Third Interview with Authorities

About six months later, he added further details, including the fact that in the days before the murder, his partner had coughed after going to places at risk for the contagion. The morning before the events he tried to drive to the region of his parents’ residence, but was intercepted by his partner and convinced to return home. On the doorstep, he hesitated to enter for more than an hour. Then, after some phone calls with his family, he came in and spent a pleasant evening with his partner. However, he sent messages to family members to give instructions on his inheritance. He had a memory lapse, a “black-out” about the events that occurred after dinner, remembering only a scene in which he tried to resuscitate his partner. It all happened because he felt followed by people with the intention of killing him, but he couldn’t say who they were. He couldn’t even say if the suicide attempt with the hairdryer was real or not. In the description of the relationship with his partner, on that occasion, he stated that they got along and that there was no jealousy. She also used to urge him to “improve professionally”.

### 3.4. Clinical and Behavioral Records from the Penitentiary Institute

Upon entering the detention facility, Mr. K was described by staff as lucid and cooperative, stereotypically reporting anxiety about possible SARS-CoV-2 infection of himself and his family. He denied hallucinations and he did not refer to thoughts or emotions about the crime. Instead, he showed a strong emotional flattening. In the following days, he began to show opposition to both psychiatric and psychological clinical interviews, also refusing the hypnotic drugs prescribed for reported insomnia. About a month after his arrival in prison, he presented an episode of aggression towards his cellmate, attempting to strangle him apparently without reason. A new episode of aggression occurred the next day, against prison officers who handed him some judicial notifications. In the following days, he had psychological sessions in which he said he was sorry for the assault on the cellmate, appearing, however, very shallow, and therapists noted the presence of a “decidedly incongruous smile” on his face.

Subsequent psychological evaluations found Mr. K always in a stubborn relational closure, suggesting an “alexithymic tendency”. Anyway, the psychiatric visits showed no psychopathological elements. In the following months, Mr. K managed to integrate effectively into the prison milieu, showing a good propensity to engage in dialogue with both other incarcerated people and with institutional figures. He had tried to improve the rehabilitation activities of other inmatesand had asked for more space and time to study by himself. His behavior changed abruptly and radically in the two weeks before the beginning of the expert interviews for the judgment of his mental state at the time of the crime. His helping attitude toward other inmates then turned into hostility and aggression, so much so that they isolated him and requested not to remain in the cell with him. Furthermore, the prison staff did not exclude the possibility that Mr. K had realized an agreement with the other inmates to show and enhance his state of mental suffering. In those days, Mr. K also assaulted a foreign inmate with a punch in the face and kicked his cellmate, always without giving a reason. Then, he spat in another inmate’s face. At the psychiatric evaluation, Mr. K was mutacic, and, while he admitted to his acts, he was unwilling to talk about it.

### 3.5. Interviews with the Experts

The first expert interview took place about twenty months after the crime. However, it ended immediately, as Mr. K reported that he did not want to talk and was “sick”.

Three months later, he was considerably more open to dialogue with the experts. From the collection of the anamnesis, it emerged that he was the youngest of five children. He reported no cases of familial mental illness and no traumatic events during his youth. In the school curriculum, there was one failure, which was overcome through the acquisition of a high school diploma and a three-year degree in nursing. He referred to having had a poor social life, despite having some friends. He said he did not particularly appreciate social situations, but remembered applying to be a student representative during the degree course. He had some emotional relationships, of which the most important was the one with the victim, while he defined the other relationships as being more “in passing”.

Except for sporadic cannabis use, he denied the use of alcohol or drugs. With respect to hi personal history of mental illness, during adolescence, he had suffered from anxiety and had had several panic attacks, for which he went to the emergency room. He usually experienced “fears” of having somatic diseases, even if he did seek reassurance from doctors. He often had frightening extrasystoles, although not continuously, especially at rest. He justified the fact that he had refused to be vaccinated against SARS-CoV-2 during imprisonment because of his fear of fatal thromboembolic adverse events. Furthermore, during the previous months, Mr. K had contracted COVID-19, albeit in a “light” form, leading him to trivialize and joke about his fear of infection. This time, he described the relationship with the victim as a situation in which they were “happy, never arguing”, stubbornly denying relational problems. They lived together after he got a job, for about two years. They met by chance in 2018 and moved in together after six months of dating. In his representation, they had a “complete affinity”, sharing shopping, motorcycles, and traveling as hobbies. She supported him professionally and encouraged him to begin a new course of study in dentistry. They were planning to get married. Regarding any jealousy problems, Mr. K stated that there had never been any quarrels with his partner over jealousy issues and that the jealousy between them was “normal”. It was their custom to check each other’s messages on their cell phones. They were also aware of each other’s personal identification number (PIN) codes. Overall, however, in Mr. K’s opinion, there was a lot of “respect” between them. The discussions began with the outbreak of the pandemic in Italy, about two months before the events. The pandemic had greatly alarmed him: he had not slept for several days, and he had felt very agitated, oppressed, and distressed, also because there was no one on the streets. Forcing himself, he continued to work as a nurse, carrying out home visits in which he wore a full-face motorcycle helmet. The feeling of oppression increased when he saw the images of military trucks carrying away the coffins of COVID-19 victims in Bergamo. Mr. K reiterated that he couldn’t explain what happened; he was “dragged away into this tunnel with no way out”. He also claimed to be a non-violent person and that he never had “problems with women”. The day before the crime he had gone to play video games with a friend, also sleeping at his house, as he needed to have some entertainment. The next day, feeling observed, persecuted, and having the need to “pull the plug”, he tried to go to his family, but at the insistence of his father and the victim, he returned home. Therefore, they showered, dined, and watched horror films, despite his unrest and anxiety for his family’s health. From that moment, his memories stopped, and Mr. K claimed to have had a “black-out”. Asked why he deleted all the text messages with the victim over the phone, he stated that he was unable to find any motivation. Concerning his aggression towards other prisoners, he stated that it was a misunderstanding, occurring when, in search of consolation, had hugged a detainee, who had become frightened.

### 3.6. Neuropsychological Evaluation

Mr. K performed assessment tests for intelligence, sincerity, and personality. The results are schematically described below:The Raven’s Standard Progressive Matrices (RSPM) [9] were administered for the measurement of non-verbal intelligence and abstract reasoning. Mr. K obtained a raw score of 34 correct answers, resulting in a position between the 25th and 50th percentile compared to a sample of the same chronological age, corresponding to a level of general intelligence that is within the norm (IQ = 95).The Wechsler Memory Scale (WMS) [10], for the evaluation of the clinically relevant aspects of memory functioning, showed a Memory Quotient equal to 105 which is generally average (M = 102.9—Ds = 5.46).The Structured Inventory of Malingered Symptomatology (SIMS) [11], in which Mr. K scored 7, which is below the cut-off score of 14, revealed no simulation attempts.The Minnesota Multiphasic Personality Inventory-2 (MMPI-2) [12] showed predominantly a ‘neurotic’ type of symptomatology, as Mr. K revealed himself to be excessively concerned for his health, with a tendency to develop physical symptoms in response to stress. Mr. K complained of sleep disturbances and somatic dysfunctions that caused him apathy and fatigue, negatively affecting his mood, and increasing his need for attention and reassurance. The reality testing was preserved although, Mr. K showed aspects of suspicion towards his milieu and resentment for what he considered an unfair trial. He also displayed immaturity, self-centeredness, and selfishness, with a need for attention and gratification from interpersonal and family relationships, with hostility towards people who did not offer enough admiration. The mood was low, but Mr. K showed that he felt able to exercise adequate control over his emotions, believing he could cope with particularly stressful situations.

### 3.7. Conclusions of the Court Expert Witness

The forensic psychiatry expert appointed by the Court concluded that Mr. K had full competence, understanding, and will, at the time of the crime. Although some clinical aspects were present at the time, such as anxiety and self-referential interpretation, these were not to be configured as a nosographically defined mental illness. Regarding the personality examination, despite the absence of a full-filled diagnosis, perfectionistic and narcissistic traits were highlighted.

### 3.8. Conclusions of the Expert Witness on the Defense Side

The defense consultant for Mr. K concluded instead by affirming a diagnosis of Brief Reactive Psychosis, and therefore complete incompetence at the time of the crime. Supporting this diagnosis were several elements, including the fact that Mr. K felt followed by the victim’s father and brother in the days immediately preceding the crime; that he felt observed; that he feared for his life and that of his family, and believed that his partner had infected him; that he was in a condition of anguish. According to the consultant, the psychotic disorder had developed on a psychic substrate given by an Obsessive–Compulsive Disorder (OCD), and at the time of the evaluation was “in a phase of good compensation”. To support the diagnosis of OCD, the consultant reported that Mr. K’s family described him as a very elegant and well-groomed person. For example, he used to go home immediately if he forgot to put on perfume before going out; he was obsessed with fashion and the order of his clothes; he was very precise on the cut of his beard, which had to be strictly symmetrical bilaterally, as he continued to do during his detention. Finally, it was to be noted that the projective tests proposed by the consultant, namely the Rorschach Test and the Thematic Apperception Test (TAT), were not evaluable due to Mr. K’s refusal to complete them.

## 4. Diagnostic Features of Brief Acute Psychotic Disorders

The central aspect of brief acute psychotic disorders, mainly BPD and ATPD, is their rapid onset, without a prodromic phase of symptoms relating to the psychotic dimension. According to DSM-5 [13], a diagnosis of BPD is possible in the presence of one or more of the following symptoms: (1) delusions, (2) hallucinations, (3) disorganized speech (e.g., frequent derailment or incoherence), and (4) grossly abnormal psychomotor behavior, including catatonia. For diagnosis, at least one of the first three symptoms must be present. To these, in defining the ATPD, the ICD-11 adds the experiences of influence, passivity, or control (see Table 1) [14]. Other symptoms can be related to confusion, both emotionally and cognitively. Furthermore, the extreme variability in the quality and intensity of symptoms is frequent, together with their polymorphism, both intra- and inter-day, as provided among the essential criteria for the ICD-11 and described in the narrative presentation of the disorder by the DSM-5. ICD-11 itself requires the necessary absence of negative symptoms (i.e., affective flattening, alogia or paucity of speech, avolition, asociality, and anhedonia).

Normally, a brief acute psychotic episode is associated with an important alteration in functioning, even if this characteristic is not explicitly requested. In the DSM-5, it is specified, in point 2, that an eventual full return to a premorbid level of functioning is associated with symptomatic remission at the end of the episode. In ICD-11, although not among the essential criteria for diagnosis, a rapid deterioration in social and occupational functioning and an equal return to premorbid functioning with symptomatic remission are included among the additional clinical features. Another important element is the association between brief acute psychotic disorders and suicidal risk [15]. As a temporal criterion, the DSM-5 requires the episode to last at least one day but less than a month, a period after which even functioning must be restored to premorbid levels. According to the ICD-11, however, the episode can develop from a non-psychotic condition within two weeks, a feature also reported by the DSM-5 in the narrative description of the diagnostic features.

Additionally, for the ICD-11, the duration of the symptoms is not expected to exceed 3 months, most commonly lasting from a few days to 1 month. The psychotic episode must not be a consequence of a somatic disorder (e.g., a brain tumor, subdural hematoma, thyrotoxicosis) or of substance use (e.g., a drug of abuse, a medication) or abstinence (e.g., alcohol withdrawal). Regarding stressors, the DSM-5 explicitly provides for specifiers on their presence or absence, while the ICD-11 merely reports, in the narrative description, that an ATPD is commonly preceded by an acute stress episode, without being a diagnostic requirement. A differential diagnosis should be made against a depressive or bipolar disorder, and other psychotic disorders such as schizophrenia, schizophreniform disorder, and catatonia. A case of malingering or factitious disorder must also be excluded, especially if there is a secondary advantage. From an epidemiological point of view, Brief Psychotic Disorder has a gender ratio of M:F equal to 1:2, with onset on average occurring during the mid-30s [13]. Predictors of a favorable outcome include acute onset, short duration, good premorbid functioning, and female gender [14].

## 5. Brief Psychotic Disorder during the COVID-19 Pandemic: A Mini-Review

The review of the literature made it possible to identify several cases of brief acute psychotic disorders related to the pandemic period. A multicenter study reported by Valdés Florido et al. [16], conducted in southern Spain (Andalucía) from March to May 2020, during a state of emergency and national confinement due to an ongoing pandemic wave, describes 33 individuals with brief psychotic episodes related to the emotional stress of the COVID-19 pandemic that met the DSM-5 criteria for “BPD with marked stressors”. Subjective stressors related to the pandemic were considered fear of infection (for themselves or loved ones), loss of a family member, domestic confinement, and occupational and socio-economic consequences. From the socio-demographic analysis of the patients, an average age of 42.33 years (SD ± 14.04, range 19–65) emerged, without significant gender differences. Most (84.8%) lived together with other people, maintaining a good premorbid psychosocial adaptation (81.8%). From the collection of the clinical history, it emerged that more than a third (36.4%) had a previous episode of BPD, while a smaller percentage had a preceding diagnosis with substance use (15.2%), depressive (9.1%) and anxiety (9.1%) disorders. A family psychiatric history was present in about one-fifth of the cases. The main symptomatic characteristics of the disorder were represented in this order: delusions in 84.8%, hallucinations in 42.4%, disorganized speech in 39.4%, and grossly disorganized or catatonic behaviors in 45.5%. Additionally, in 45.5% of cases, first-rank symptoms of schizophrenia were present, while in 57.6% the psychotic issues concerned the pandemic. Approximately one-quarter of the cases (24.2%) had suicidal symptoms. For timing, there was an abrupt onset (<48 h) in 42.4% of cases, a median duration of untreated psychosis (DUP) of 5 days (IQR 3.75–11.5), and a symptomatic remission achieved on average in 15 days (IQR 7.75–25.75). Most patients (84.8%) required hospitalization. Notably, none of the patients tested positive for SARS-CoV-2. D’Agostino et al. [17] described six cases (three males and three females) of patients with BPD with marked stressors hospitalized in Italy during the lockdown period, in the week between 25 April and 2 May 2020. The main stressors reported by the whole sample concerned both the intense fear of contagion and domestic confinement. Additionally, three cases reported other major stressful life events in the previous 12 months. None tested positive for SARS-CoV-2 and no significant somatic diseases emerged except for mild cortical atrophy in one case. There were no personality disorders in any of the cases, nor previous psychiatric disorders. Psychosocial functioning was also normal. Quickly after the beginning of the restrictive measures to control the contagion, they had begun to develop delusional themes with prevalently paranoid, nihilistic, or mystical–religious themes. In most cases, auditory hallucinations, including imperative or visual hallucinations, bizarre and disorganized behavior, and aggressivity against themselves and/or others were associated. The treatment was mostly based on the use of antipsychotics in medium-low dosages and, in some cases of depressed mood, with the addition of antidepressants. A peculiarity of the cases also consisted in the good efficacy of the treatments, with reasonable rapidity, as well as in the full recovery of psycho-social functionality. In addition, in half of the cases, drug dosage was reduced to a low dose of antipsychotics, and in the other half, therapy was completely suspended, with no case experiencing a symptomatic relapse or functional deterioration. Similarities were reported in four cases further described by Valdés-Florido et al. [18], which occurred in the first two weeks of the national lockdown in Spain in 2020. All of them developed delusional disorders very rapidly, although only one had a history of previous psychotic disorders. In half of the cases, suicidal behavior was also observed. In contrast, Huarcaya-Victoria et al. [19] reported a case of auditory hallucinations developed in a condition of marked anxiety related to COVID-19. It was a case of a woman with no psychiatric history, who, after a dental treatment conducted by an operator without a protective mask, began to worry about having contracted the infection and to become increasingly anxious. She also presented with malaise and insomnia, and 15 days after the event, she began to hear a voice ordering her to go to a medical center to check for infection. The clinical presentation worsened further in the following days, with the appearance of delusional themes of demonic possession and imperative hallucinations that ordered her to kill her family. Hospitalized, she appeared confused, disoriented in time and space, with little insight, and with both auditory and visual hallucinations, delusional themes of persecution, possession, guilt, and punishment, among other symptoms. An interesting case is also the one reported by Marouda et al. [20] about a young man, also without a personal or family psychiatric history. The only noticeable risk factor was the occasional consumption of cannabis, with the last intake occurring about ten days before admission. The involuntary hospitalization took place after he had begun to feel agitated for two days and had presented megalomanic, persecutory, and bizarre delusions. He became aggressive towards those he believed members of a “bad group”, while he felt engaged in defending the “good group”, saving them from the viral threat. In this case, too, there were themes of demonic possession in the form of animal presence in the patient’s body. After treatment with high-dose antipsychotics, the psychotic symptoms and aggression significantly and very rapidly reduced. Further similar cases were reported by other authors in several countries worldwide [21,22,23,24,25,26]. In all reported cases, psychotic symptoms were constantly present in the form of delusions with various themes but always involved SARS-CoV-2 infection. To a less degree, delusions were accompanied by hallucinations, bizarre, cognitive, and associative alterations, insomnia, hyporexia, dysphoria, and suicidal behavior. However, in none of the cases reported was the transition to a particularly violent act with relative injury or death of the victim.

## 6. Discussion

### Explanatory Hypothesis about Mr. K’s Case

The diagnosis of brief acute psychotic disorders, such as BPD, influencing the evaluation of criminal responsibility, represents an important issue in the context of forensic psychiatry and is a frequent cause of controversy among experts. This occurs particularly in countries, such as Italy, where the law prohibits the condemnation of people with diminished capacity at the time of the crime, a mental state that an acute psychosis could cause [27]. Moreover, the cases reported in the literature were not associated with violent crimes. The particularity of the pandemic context, with its pathoplastic potential, represents a further challenge for the definition and contextualization of the features of brief acute psychotic disorders. In this particular regard, realizing a mini-review of the literature on cases of COVID-19-related psychosis was very helpful. The peculiar characteristics that emerged about clinical presentation were compared with those of the reported case so that it was possible to clearly show that they were very different. In the latter, there were no perceptual disturbances and delusions that required immediate hospitalization and the use of antipsychotic therapy, the only way to achieve remission and return to a premorbid level of functioning. Furthermore, there were no symptomatologic or behavioral fluctuations, while they were particularly evident in the case of Mr. K. Our hypothesis is that the crime involving Mr. K was based on his predominantly narcissistic, albeit very hidden, personality structure. The association between narcissistic personality traits and violent behavior has emerged as being significant in some studies [28], in particular when they configure as delusions of a grandiose nature, elation, and anger [29,30]. In Mr. K, narcissistic and partly antisocial traits could be assumed, for example, from what emerged from records of the Penitentiary Institute. He trivialized the infection by SARS-CoV-2 about which he had had so much worry; he had started to act personally, interfacing with the Direction, with recreational and rehabilitative activities, making himself pleasant in the eyes of the other cellmates; this representation underwent a rapid twist in the period before the expert’s assessment, when Mr. K was the author of a series of aggressions against other inmates until transferred from the institution; during the interviews, he minimized these attitudes, suggesting that there had always been misunderstandings from others. Generally, attitudes of concern for the improvement of the conditions of others soon seemed to be very false. The rapid change in behavior had even made the Direction of the Institute hypothesize that Mr. K had somehow manipulated the other inmates to obtain benefits, such as recognition of mental illness or psychological distress.

The biography of Mr. K also showed continuous identity fibrillation, which could be indicative of personality conflicts. He described himself as a somewhat lonely person, but later reported that he had applied as a student representative during the degree course. He did not consider himself a jealous and possessive person, but then revealed that he checked the victim’s phone and had her security code. In addition, he had met the victim in a healthcare setting, where he practiced the profession of a nurse while she was a medical student. Although it is not possible to know any conscious or unconscious conflicts related to this disparity of roles, given his closure to dialogue, the emotional superficiality of Mr. K, and his opposition to projective tests, some suggestive facts emerged. The victim would have graduated shortly after the tragic event, thus becoming a doctor; she urged him to “improve professionally”; he intended to try a new course of study in dentistry, a course which was, however, very difficult in terms of access, due to the limited number of places, and subsequently to complete. All this could have made Mr. K feel forced into an inferior condition with respect to his partner, resulting in a narcissistic wound. Mr. K then described his relationship with the victim in idealistic, superficial, and, again, contradictory terms: they always had a peaceful relationship, except for having “the quarrels of all the couples”. The main qualities he recognized in the victim were those of sharing the same interests, namely shopping and motorcycles. Another fact is also significant, whereby the day before the murder, he spent the evening with a friend playing video games, spending the night at his home. Mr. K did not find this behavior incongruent with the reported state of anguish for the pandemic, highlighting how the aggressiveness was polarized exclusively at the victim. The latter, indifferently together with her family, represented for all intents and purposes a threat to the physical and mental existence of Mr. K and his family. Mr. K admitted that the relationship with the victim was his first “emotionally” important relationship, while previously he had only short-lasting affairs. It is reasonable to think that such affairs had the effect of sustaining and stabilizing Mr. K’s self-esteem, which was significantly undermined by a prolonged relationship, resulting in him being intellectually, emotionally, and professionally unbalanced, while nevertheless initially contributing to the building up of a grandiose false Self, while keeping a deep sense of envy under the skin [31]. The pandemic, as an unexpected and existentially overwhelming event, probably contributed to deconstructing Mr. K’s artifical existence and revealing to him the emptiness of his inner world. This intolerable condition led him to need to run away, back to a place of family undifferentiation. At attempts by the victim to stop him, the aggressive compulsion found its fatal objective. According to Nancy McWilliams, people with narcissistic structures have a deep fear of the fragmentation of their Self and, unable to be aware of it, shift it to a somatic concern, indulging in hypochondriac worries and morbid fears of dying [32]. Finally, Mr. K never declared any sense of despair over the death of his partner, whom, moreover, he was to have married soon after, and with whom he was part of a happy couple, he said. Instead, he stated that if she had not stopped him that day, all this would not have happened. The inability to feel remorse and pity for a broken young life seals the definition of a narcissistic personality with a high level of psychopathy [33]. This combination is particularly important considering that narcissism and psychopathic indifference toward the victims significantly increases the risk of serious violence [34]. Concerning the aspects of extreme care of his appearance, indicated as characteristics of OCD by the defense consultant, it is evident that these are rather attributable to a personality narcissistically obsessed with appearances.

## 7. Conclusions

The case presented highlighted the importance, in the context of psychiatric-forensic evaluation, of integrating the interviews with the offenders with the careful collection of biographical information, the neuropsychological assessments, and the additional diachronic observations provided by staff who interface with them. However, in some peculiar conditions, this may not be enough. This is the case, for example, for the COVID-19 pandemic, which has created a totally new and unexpected socio-health scenario. In this context, many difficulties have arisen in distinguishing whether the environmental situation represented an efficient cause or only an epiphenomenon for the symptomatology of the patients. In the case presented here, the review of the literature provided valuable information for comparing the clinical picture of the person examined with those of the other case reports. In conclusion, it is advisable to base the forensic assessment on multifaceted aspects, through the analysis of multiple data sources to have a framework that should be as comprehensive as possible. The importance is increasingly evident of combining different fields of knowledge, as well as using standardized assessment tools [35] to arrive at valid medico-legal conclusions and shared programs between the judiciary and social and mental health services. This has been well demonstrated by the experiences of the creation of liaison services to provide alternatives to imprisonment for criminals [36]. Discerning forensics from health cases represents an important issue of risk management. This allows for the avoidance of the consumption of economic and professional resources, available for patients, avoids unnecessary risks for healthcare personnel, and reduces the impact of defensive medicine on clinical outcomes [37,38,39,40].

## 8. Limits

The main limitations of our study are represented by the report of a single case and the comparison in terms of clinical features with the other cases reviewed. Furthermore, despite the usefulness of the assessment tests, the diagnostic conclusions were based on the decision of the court expert witness, that is, not standardized. Regarding the proposed tests, it is worth noting that not all of them were completed, namely the projective-type tests. Finally, an important limitation is intrinsically connected to the forensic context, such as a situation where the subject is aware of the legal implications of his statements, with obvious repercussions on the relationship with the evaluating experts.

## 9. Implications

Despite the limitations, we believe that our study can be useful in several regards. It could stimulate the sharing of significant experiences in the field of forensic psychiatry, including in narrative terms, considering that documents for Courts are mainly of this type. The comparison of experiences in this context has become fundamental during the SARS-CoV-2 pandemic, which is a completely new and unexpected situation determining numerous repercussions on the mental health of the general population and specific subgroups. Specifically, our case presented diagnostic difficulties due to a problematic historical contextualization, with possible confounding factors, as was highlighted by the comparison with the other cases reviewed. The field of research in forensic psychiatry is still rather anchored to traditional research methods, borrowed from clinical research. Given the complexity of this field, it could be interesting to introduce new means for research and experience sharing (e.g., through online qualitative research tools [41,42]). A further point is represented by the scarcity of forensic research on personality disorders. Such disorders, well known in psychological and psychiatric contexts, have received poor consideration in the forensic field, except when associated with severe psychopathology. Both psychological and forensic psychiatric research should focus on these disorders, especially with regard to their repercussions regarding inter-individual violence, such as intimate partner violence, where personality traits can be expressed in a highly lethal manner.

## Figures and Tables

**Table 1 ijerph-19-14826-t001:** Brief acute psychotic disorders diagnostic configuration as BPD and ATPD.

Diagnosis	BPD *	ATPD **
**Symptoms**		
Delusions	+ ^#^	+
Hallucinations	+ ^#^	+
Disorganized speech	+ ^#^	+
Grossly disorganized or catatonic behavior	+	+
Experiences of influence, passivity or control	-	+
Lack of a prodrome period	-	+
Infra-day or inter-days symptoms rapidly change, both in nature and intensity	-	+
Absence of negative symptoms	-	+
**Time**		
Duration of an episode of the disturbance is at least 1 day but less than 1 month	+	-
Progression from a non-psychotic state to an evident psychotic state within 2 weeks, with a duration ≤ 3 months (commonly lasts from a few days to 1 month)	-	+
Etiology unrelated to medical condition, substance or medication, including withdrawal effects	+	+
**Differential Diagnosis**		
Not better explained by Major Depressive or Bipolar Disorder with psychotic features	+	-
Not better explained by Schizophrenia or other primary psychotic disorders	+	+
Absence of culturally sanctioned response patterns	+	-
**Functioning**		
Usually associated with a rapid deterioration in social and occupational functioning (Additional Clinical Features)	-	+
Eventual full return to premorbid level of functioning at symptomatic remission	+	+
**Specifiers**		
With marked stressor(s)	+	-
Without marked stressor(s):	+	-
With postpartum onset	+	-
With catatonia	+	-
Course specifiers	+	+

* A: DSM-5, Brief Psychotic Disorder diagnostic criteria [298.8]; ** B: ICD-11, Acute and Transient Psychotic Disorder diagnostic criteria [6A23]; ‘+’ diagnostic criterion is present in this diagnostic system; ‘-’ diagnostic criterion is not present in this diagnostic system. ‘^#^’ The presence of at least one of the first 3 symptoms listed is required to make a diagnosis of Brief Psychotic Disorder according to DSM-5.

## Data Availability

Not applicable.

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
