# Peer review of "When COVID-19 Is Not All: Femicide Conducted by a Murderer with a Narcissistic Personality “Masked” by a Brief Psychotic Disorder, with a Mini-Review"

_ijerph, 2022, doi:10.3390/ijerph192214826_

Round 1

Reviewer 1 Report

I have enjoyed reading the paper and please see my attached feedback. I would make sure you ask at least 1-2 friends reading the paper thoroughly to make sure it is at a much better level, especially for minor language issues.

Author Response

Dear reviewer,
We want to thank you for your feedback. It's a very pleasure for us if you have appreciated our work.
Following your suggestions, we have corrected some linguistic imperfections and we have included the sections "Limitations" and "Implications", adding a note on the usefulness of new tools for research, such as Online Photovoice.
Thank you, sincerely.

Reviewer 2 Report

A very interesting article. I would like to thank the authors for their insightful presentation of the results of medical and psychological research as well as data from court proceedings. The authors rightly note that it is very difficult to distinguish between a narcissistic personality with psychopathological features and psychotic disorders. An additional challenge in the diagnosis process is to take into account circumstances that are completely new and we know little about their impact on the functioning of people.An example is the COVID-19 epidemic.

Presenting your own diagnostic procedure and the cases described in the literature in order to better understand the patient and make a more accurate diagnosis is by all means beneficial. It brings us closer to the development of useful operating procedures. Congratulations to the authors.

I propose to amend the description of the patient's results in such a way that point 3.5 is not a separate paragraph. Now this paragraph only contains two sentences. I suggest combining it with the next paragraph.

Author Response

Dear reviewer,
We are very pleased with your congratulations, and we want to thank you for your feedback.
As you correctly suggested, we combined paragraphs 3.5 and 3.6, under the title “Interviews with the experts”.
Thank you, sincerely.

Reviewer 3 Report

Dear authors, 

Thanks for the opportunity to read your work. In the present study a feminicide case occurred in Italy during the first phase of covid-19 pandemic is analysed in order to clarify discrepancies between forensic evaluations. Despite its originality there are some aspects that need to be improved. 

Abstract: I think it should be better stated the scientific method used to analyse the case and to carry out the literature review. What tools? What hypothesis? What results?

Introduction: beyond to the contextualization of the historical moment at which this feminicide occurred, it would be important to include a national and international literature review on the phenomenon under study or analyses of similar cases during the pandemic period. Moreover, it would be good to deepen the discretion of the use of risk assessment tools in forensic contexts compared to the international panorama. In this regard I point out some articles: 

-       Sorge A, Borrelli G, Saita E, Perrella R. Violence Risk Assessment and Risk Management: Case-Study of Filicide in an Italian Woman. Int J Environ Res Public Health. 2022 Jun 7;19(12):6967. doi: 10.3390/ijerph19126967. PMID: 35742216; PMCID: PMC9223206.

-       Mazza, M.; Marano, G.; Lai, C.; Janiri, L.; Sani, G. Danger in danger: Interpersonal violence during COVID-19 quarantine. Psychiatry Res. 2020, 289, 113046.

-       Piquero, A.R.; Jennings,W.G.; Jemison, E.; Kaukinen, C.; Knaul, F.M. Domestic violence during the COVID-19 pandemic-Evidence from a systematic review and meta-analysis. J. Crim. Justice 2021, 74, 101806

-       Barchielli, B.; Baldi, M.; Paoli, E.; Roma, P.; Ferracuti, S.; Napoli, C.; Giannini, A.M.; Lausi, G. When “Stay at Home” Can Be Dangerous: Data on Domestic Violence in Italy during COVID-19 Lockdown. Int. J. Environ. Res. Public Health 2021, 18, 8948.

-       Lausi, G.; Pizzo, A.; Cricenti, C.; Baldi, M.; Desiderio, R.; Giannini, A.M.; Mari, E. Intimate Partner Violence during the COVID-19 Pandemic: A Review of the Phenomenon from Victims’ and Help Professionals’ Perspectives. Int. J. Environ. Res. Public Health 2021, 18, 6204. 

Materials and methods: what are meant by mini review? What are the methodological steps that could allow other researchers to replicate it? What keywords? Define precisely the time period and the inclusion criteria used.

The case of Mr. K: in the first two paragraphs of this section there are repeats. 

What about ethical committee? What about permission to access documentation?

Results: By including a review of literature on the phenomenon of feminicides during the pandemic, it will be possible to comment more extensively on the results in the discussion section.

An English revision is appropriate. 

Kind regards 

Author Response

Dear reviewer,
Firstly, we want to thank you for your feedback and suggestions.
Consequently, we have made the following changes to our paper:
- Abstract: we reported the results of the literature review, and we remarked on the difference with our case.
- Introduction: reviewing the articles you suggested, we added information about IPV in the introduction and we used other useful information from those articles to enrich the discussions and the conclusions.
- Materials and methods: we added the search terms we used, defining explicitly the period and the inclusion criteria used. We have not explained the meaning of "Mini-review" as it is a specific type of article provided by the journal.
- We have eliminated the repetitions in the first two paragraphs of this section “The case of Mr. K”.
- We specified that the documentation was obtained from the public legal debate.
- We enriched conclusions with more specific references on the phenomenon of IPV and the usefulness of risk assessment instruments.
- Finally, we revised the English language
Thank you, sincerely.

Round 2

Reviewer 3 Report

Dear authors, 

I believe the article has improved as a result of the changes you have made, but I still have doubts about the ethical issues.

What does it mean that the information examined comes from the public debate?

There are very detailed information so I think it is worth specifying their source and who allowed the study and publication.

Kind regards 

Author Response

Dear reviewer,
Thank you for your suggestions. We have made further changes to clarify the ethical terms of our work. In particular, we have explained how the material reported by our work, regarding a forensic case already judged by the Judicial Authority, is in the public domain and freely available. In this way, full respect for the privacy of the person is provided. 
Best regards
